# Molecular Dynamics Simulations of Drug-Conjugated Cell-Penetrating Peptides

**DOI:** 10.3390/ph16091251

**Published:** 2023-09-05

**Authors:** Márton Ivánczi, Balázs Balogh, Loretta Kis, István Mándity

**Affiliations:** 1Institute of Organic Chemistry, Semmelweis University, Hőgyes Endre Utca 7., H-1092 Budapest, Hungarykis.loretta97@gmail.com (L.K.); 2Artificial Transporters Research Group, Institute of Materials and Environmental Chemistry, Research Centre for Natural Sciences, Magyar Tudósok Körútja 2., H-1117 Budapest, Hungary

**Keywords:** cell-penetrating peptides, molecular dynamics, drug conjugates, biological membrane, penetratin, cyclic peptides, explicit membrane model, intracellular target, Desmond, in silico simulation

## Abstract

Cell-penetrating peptides (CPPs) are small peptides capable of translocating through biological membranes carrying various attached cargo into cells and even into the nucleus. They may also participate in transcellular transport. Our in silico study intends to model several peptides and their conjugates. We have selected three CPPs with a linear backbone, including penetratin, a naturally occurring oligopeptide; two of its modified sequence analogues (6,14-Phe-penetratin and dodeca-penetratin); and three natural CPPs with a cyclic backbone: Kalata B1, the Sunflower trypsin inhibitor 1 (SFT1), and Momordica cochinchinensis trypsin inhibitor II (MCoTI-II). We have also built conjugates with the small-molecule drug compounds doxorubicin, zidovudine, and rasagiline for each peptide. Molecular dynamics (MD) simulations were carried out with explicit membrane models. The analysis of the trajectories showed that the interaction of penetratin with the membrane led to spectacular rearrangements in the secondary structure of the peptide, while cyclic peptides remained unchanged due to their high conformational stability. Membrane–peptide and membrane–conjugate interactions have been identified and compared. Taking into account well-known examples from the literature, our simulations demonstrated the utility of computational methods for CPP complexes, and they may contribute to a better understanding of the mechanism of penetration, which could serve as the basis for delivering conjugated drug molecules to their intracellular targets.

## 1. Introduction

Membrane-active peptides are divided into two main categories: antimicrobial peptides (AMPs) and cell-penetrating peptides (CPPs). The AMPs (also known as host defense peptides or HDPs) are part of the innate immune response as potent, broad-spectrum antibiotics acting through the destabilization of membranes of the pathogens. Unlike AMPs, CPPs can translocate into living cells and their organelles without lasting damage under physiological conditions [1,2,3]. This peculiarity makes CPPs adequate to deliver drugs and other compounds to intracellular targets or across the blood–brain barrier [4,5].

Conjugated CPPs could improve various pharmacokinetic properties of drugs, including, for example, poor delivery and low bioavailability, and they may also decrease toxicity. Nearly all types of ‘cargo’ can be transported: small molecules, diagnostics, macromolecules (DNA, RNA, antibodies, peptides), or even nanoparticles [4]. The conjugation of penetratin with paclitaxel was one of the first applications. It not only increased the solubility of the compound but also helped to reach the nucleus, thereby improving the anticancer activity of the compound [6]. One of the most recent successful applications of CPPs is the development of orally active insulin with preclinical development in 2021 [7]. Penetratin also seems to be capable of delivering isoniazid into mycobacteria [8].

Different strategies were published in the literature for the conjugation of the cargo; for example, certain macromolecules could be attached in a non-covalent manner via charge-dependent complex formation with the CPP. However, small molecules were mostly bound covalently [4]. Conjugation can be formed by either the terminals of the peptide or through a side chain with the application of an appropriate functional group. A linker is frequently used in order to increase the distance between the peptide and the active ingredient or providing reversibility (capability of detachment under appropriate conditions) [9,10,11,12,13]. CPPs also can be the component of more complex drug delivery systems combined with polymers, dendrimers, or antibodies for targeting, especially in cancer therapy [14,15]. Homing peptides and targeting ligands are capable of combining with CPPs to improve cell specificity [16].

Several possible mechanisms of the penetration are described in the literature, with the two main types being energy-dependent or energy-independent [17,18]. The energy-dependent mechanism indicates that contribution by the cell is needed via nearly all ways of endocytosis: phagocytosis, macropinocytosis, clathrin-coated vesicles, caveola formation, and constitutive endocytosis (Figure 1A–E). Energy-independent penetrations may occur spontaneously through direct translocation, membrane thinning, pore formation, and inverted micelles (Figure 1F–J). According to the carpet model mechanism (Figure 1K), the peptides in high concentration are adsorbed on the surface of the membrane, then the lipids form toroid aggregates stabilized by the amphipathic peptides. This causes serious damage leading to cell death, unlike any other mechanisms belonging to this category [13,19]. Some peptides can penetrate and form pores even as oligomer complexes [20].

Our aim is to clarify the possible mechanisms with molecular dynamics simulations through a number of examples. The original aim was to model the internalization process, but it did not occur. Therefore, we focused on some aspects that might be involved in the penetration instead, such as the formation of intermolecular interactions, orientation, and conformational changes of the penetratin and analogues, and the influence of the conjugated small molecule on the process. In the literature, a wide range of computational chemistry techniques were used to investigate CPPs. Several attempts have been made to predict penetrating ability, including cheminformatic filters, artificial intelligence-based models, and quantitative structure–property relationships. In addition, a number of studies were aimed at the modeling of the mechanism of penetration, applying molecular dynamics [21,22,23,24,25,26].

The role of the membrane potential of living cells seems to be important in most mechanisms [27]. The correlation between CPPs and transmembrane potency may indicate that generating positive charges on the outer surface of the plasma membrane could decrease the free energy barrier associated with translocation. Additionally, it triggers pore formation [28]. It appears that the possibility, speed, and mechanism of penetration through asymmetric membranes may not be the same [29]. In the case of arginine-rich peptides, the penetrating capability may correlate with the backbone rigidity [30].

Lensink et al. carried out one of the first studies on simulating penetratin in 2005. The peptide did not translocate during the simulation utilizing Gromacs, but several interactions between the peptide and the membrane lipids were observed [31].

Herce and Garcia published an important study about MD simulations of CPPs in 2007. The HIV Tat peptide was found to translocate spontaneously, mainly via transient pores, while the positively charged side chains interacted with the phosphate groups of the membrane lipids [32]. Their next article in 2009 was about pore formation by arginine-rich peptides [33]. According to simulations and experiments, guanidium groups can lead the penetration not only in peptides, but in other similar macromolecules too [34]. The mentioned interactions between Tat and membranes were experimentally confirmed using X-ray diffraction [35,36].

In the following section, a few examples with respect to the topic of CPP penetration with MD will be addressed. The role of the membrane tension was confirmed in simulations with the coarse-grained MD method, with the finding that polyarginines in low concentrations were only adsorbed on the membrane surface, whereas translocation in higher concentration was completed in less than 100 ns [37].

In a study by Bennett in 2016, the CM15 antimicrobial peptide was shown not to translocate, but it only entered the membrane and reached its equilibrium point inside the lipid bilayer [38]. Further studies applying a similar setup makes the pore formation likely [29,39].

The direct translocation of pVEC (amphipathic CPP) was successfully simulated using the steered MD method. The penetration was led by the N-terminal amino acid of the peptide, while the cationic side chains were interacting with the phosphatide groups, enhancing the adsorption on the membrane [40]. The importance of transmembrane electric potential was also demonstrated in silico by the MARTINI coarse-grained MD and metadynamics simulations, in which the translocation of arginine-rich design peptides were successfully promoted by the introduction of the electrostatic gradient [41].

Ulmschneider, in 2017 and 2018, investigated the mechanism of antimicrobial peptides using molecular dynamics simulations [42,43]. Again, the importance of arginine was confirmed [44]. In the case of hydrophilic peptides, the computed free energy of membrane insertion does not depend on the MD method [45]. The energetic aspect of the transmembrane penetration of peptides was studied by Yao et al. in 2019 [46].

MD simulations were used to validate and prioritize the penetration of CPPs generated by artificial intelligence, and a novel CPP sequence named Pep-MD was de novo identified and then synthesized. Later, its penetration potential into living cells was demonstrated by in vitro experiments [47].

In simulations of CPPs containing unnatural amino acids, the mechanism of penetration may depend on the lipid composition of the membrane. In one study by Gimenez-Dejoz et al., the methyl groups of α-aminoisobutyric acid facilitated hydrophobic interactions inside the membrane, while side chains of lysines formed electrostatic interactions with the phosphatide groups in the outer layers. Other components of the membrane may also influence the penetration. In the same study, the addition of cholesterol into the bilayer decreased the efficiency of CPPs [48].

In some of the above-mentioned examples, coarse-grained models were used because of their cost-effectiveness in the case of limited computing capacity. However, the drastic improvement in computational capacities allowed for applying all-atom calculations instead of the simplified coarse-grained models when both explicit solvent and membrane models could be included [49]. Therefore, all-atom MD simulations have been applied in our study to investigate the CPP membrane contacts and clashes as well as their changes over time. We assume that this approach is suitable for the modeling of direct penetration, but its applicability might be limited for other energy-independent mechanisms [50,51].

In the current study, three linear and three cyclic CPPs were selected, representing two significantly different groups of CPPs. The linear ones were penetratin and its two known analogues (Table 1). Penetratin is one of the best known CPPs, and has been included in numerous studies; therefore, it is an ideal reference molecule. The two modified analogues (6,14-Phe-penetratin and Dodeca-penetratin) are lesser known, but their membrane translocation capabilities have been established in vitro. In the first analogue, the replacement of tryptophans with phenylalanines showed weaker penetration in vitro. In the dodeca analogue, in contrast, the removal of one cationic amino acid (together with three more) did not affect penetration. We intended to investigate whether these small differences would affect our simulations and learn if we would be able to differentiate between them. The three cyclic CPPs (Kalata B1, Sunflower trypsin inhibitor 1 (SFTI-1), and Momordica comhinchinensis trypsin inhibitor II (MCoTI-II)) (Table 1), although also known as CPPs, have been less examined. With the inclusion of these inhibitors, we intended to investigate whether the elimination of the charged N- and C-termini of the chain (as a result of the cyclization) and conjugation through the side chain (instead of the N-terminal) would affect the simulation. Furthermore, these peptides have various sizes, and consequently, we were able to analyze the penetration of peptides with small, medium, and large sizes.

Penetratin is a fragment of Antennapedia homeoprotein (helix III) isolated from Drosophila melanogaster and is one of the most frequently investigated CPPs. It consists of sixteen amino acids, including seven amino acids with cationic (three arginines, four lysines) and three with aromatic side chains (two triptofanes and a phenylalanine). The secondary structure of penetratin is roughly helical. However, depending on the conditions, it can be either α-helix or 3_10_-helix (Figure 2A). Its capability of spontaneous translocation through cell membranes has been experimentally certified [56,57].

6,14-Phe-penetratin is an altered version of penetratin, in which both tryptophan units have been replaced with phenylalanines, resulting in a mostly α-helical and less flexible conformation compared to that of penetratin (Figure 2B). As a consequence, the biological activity was much lower than that of the original peptide, yet it was still a functional CPP [56,57].

Dodeca-penetratin is another modified version of penetratin, built of only 12 amino acids instead of 16 (Figure 2C). Even with this change, it has been shown to be effective because the critical cationic and aromatic residues have remained, despite its conformational instability [56].

Kalata B1 is a member of the cyclotide family isolated from the plant Oldenlandia affinis. Beyond its capability of membrane penetration, it has been characterized by high chemical and thermal stability together with pharmaceutical and insecticidal properties. Its 29 amino acids form a long cyclic backbone resulting from the formation of a peptide bond between the N- and C-terminals of the chain. This already hindered structure is further stabilized by three disulfide bonds formed within the peptide called knot motif, making it even more rigid and stable (Figure 2D) [53].

SFTI-1 is another cyclic CPP of natural origin (isolated from Helianthus annuus), with the sunflower trypsin inhibitor indicating its enzymatic function. It is built of 14 amino acids, and its conformation is characterized by two anti-parallel β-strands stabilized by seven hydrogen bonding and a single disulfide bridge (Figure 2E) [54].

MCoTI-II (isolated from Momordica Cochinchinensis) is another example of a macrocyclic knotin with a similar enzymatic function. The group it belongs to was named squash trypsin inhibitors. It is made of 34 amino acids, and its structure is stabilized by three disulfide bonds (Figure 2F) [55].

In the current study, three different types of drugs with known penetration-related difficulties were selected with different lipophylicity values (Table 2). All three were known for their peptide conjugates in the literature, but only one of these (rasagiline) was investigated before as a complex.

Doxorubicin is a topoisomerase-2 inhibitor anticancer drug. In animal tests, the peptide-conjugated form of doxorubicin has been excreted much more slowly, and, therefore, a much lower blood concentration was needed to have an equal therapeutic effect [11].

Rasagiline is a specific, irreversible MAO-B inhibitor used for the treatment of Parkinson’s disease; that is, it has to get through the blood–brain barrier. It has been experimentally confirmed that the drug attached to the CPP was more effective than its unconjugated form [12,58].

Zidovudine, a reverse transcriptase inhibitor antiviral drug, has been developed to cure HIV infection. The peptide conjugation can increase its specificity towards the infected cells, thereby reducing the side effects [10].

**Table 2 pharmaceuticals-16-01251-t002:** Partitioning of the investigated drugs.

Drug	Molar Weight (g/mol)	Experimental logP
Doxorubicin	543.52	0.32 [59]
Zidovudine	267.24	0.04 [60]
Rasagiline	171.24	2.462 [61]

Despite the high number of papers related to CPPs, no MD simulation studies of drug-conjugated CPPs have been published, and studies based on the comparison of more different peptides in conjugation with one or more small molecules are also rare.

## 2. Results

A complete structural rearrangement was observed with penetratin in the proximity of the surface of the POPC membrane model. At first, the helix uncoiled and ceased to exist entirely, and then slowly transformed into two-strand antiparallel β-sheets connected by a β-turn, laid to the surface of the membrane. The analysis of the last frame of the trajectory also revealed that four salt bridges and eleven hydrogen bonds were formed between the peptide and the membrane molecules, whereas no π-cation interactions were observed (see Figure 3A, Figure 4 and Figure 5 and summaries in numbers in Table 3 Entry 1). Both 6,14-Phe-penetratin and dodeca-penetratin preserved their helical structure until the end of the simulations, with their N-terminal partially sinking into the POPC membrane. The axis of the 6,14-Phe-penetratin closed at about a 60° angle with the plane of the membrane, while dodeca-penetratin was almost perpendicular (Figure 3B,C, Figure 4 and Figure 5). Despite the limited area of contact (compared to those of the penetration), 6,14-Phe-penetratin formed five salt bridges, seven hydrogen bonds, and a single π-cation interaction with membrane molecules (Table 3 Entry 5). Dodeca-penetratin connected even more loosely to the membrane surface with only one salt bridge and six hydrogen bonds (Table 3 Entry 9).

During 1000 ns simulations, the cargo molecules significantly affected the position of the conjugate relative to the membrane, and in the case of penetratin, they affected the conformation of the peptide as well. In contrast to native penetratin peptide, the unfolding mentioned above was not observed in the conjugated ones. In the case of the penetratin analogues, the helical structure remained intact similar to their unconjugated counterparts. We also observed that, unlike the unsubstituted penetratin and analogues, not all conjugated peptides positioned with their terminal towards the membrane with their longer axis perpendicular or at a closing angle, and they partially sank into the membrane. The three non-cyclic doxorubicin conjugates positioned differently. Penetratin-doxorubicin was one of the two conjugates that moved away from the membrane without forming any interaction (Figure 6A and Figure 7; see the supplement for interaction diagrams that are not included in the text). The 6,14-Phe-penetratin-conjugate anchored to the membrane through its N-terminal with the doxorubicin tightly bound on the surface with three salt bridges and eight hydrogen bonds (Figure 6B, Table 3 Entry 6). The dodeca-penetratin conjugate anchored to the surface of the membrane with its C-terminal through a number of interactions with doxorubicin orientated into the opposing direction towards the water box (Figure 6C, Table 3 Entry 10). The rasagiline-conjugated penetratin and analogues always positioned with their N-terminals toward the membrane with the cargo compound sank into the bilayer. Their positions were stabilized by the formation of several hydrogen bonds and a few salt bridges between the peptide and membrane molecules (Figure 6D–F; Table 3 Entries 3, 7, 10). The simulations with the zidovudine-linear peptide conjugates showed, among the three drugs, that this compound seemed to be the least likely to penetrate, and its conjugates had significantly less interaction with the membrane. The native penetratin zidovudine conjugate was positioned with its C-terminal towards the membrane connected with only a single hydrogen bond and the N-terminal with the cargo pointing towards the opposite direction (Figure 6G, Table 3 Entry 4). The 6,14-Phe-penetratin-zidovudine was the other example with the conjugate entirely moving away from the membrane without any possible bond formation (Figure 6H, Table 3 Entry 8). Only the dodeca-penetratin-zidovudine conjugate turned with its N-terminal towards the membrane with the formation of two salt bridges, with the cargo wedged between the peptide and the membrane (Figure 6I, Table 3 Entry 12).

The structure of the unconjugated cyclic peptides did not show any significant change during the 1000 ns simulations (Figure 8). In the course of the runs, they all positioned toward the membrane and then tightly adhered to its surface with minimal sinking into the bilayer. Unlike penetratin, these cyclic peptides—with the exception of MCoTI-II—have only a few amino acids with polar side chains, which limits their capability to form ionic interactions. A high number of mostly uncharged hydrogen bonds were observed, where the peptide heteroatoms were the donors and the heteroatoms of the membrane were the acceptors. The lack of aromatic side chains also excluded the formation of π-cation interactions with the positively charged choline groups of the POPC membrane. The overall impact of conjugation in the case of the cyclic CPPs was much less significant than those of penetratin and its analogues. At the end of the simulation, the cyclic peptides had fewer interactions with the membrane compared to penetratin and its analogues. A possible explanation is that penetratins were mostly positioned outside the membrane where the polar phosphorous groups were available to form H-bonds and salt bridges. In contrast, the cyclic peptides sank into the hydrophobic interior of the membrane more deeply, further away from the polar surface. However, there were some exceptions, such as MCoTI-II-doxorubicin and MCoTI-II-zidovudine conjugates, both with a significant number of hydrogen bonds and salt bridges (Figure 9, Table 3 Entries 22 and 24).

At the end of the 1000 ns simulations, the positions of all cyclic conjugates compared to the POPC membrane model were very similar to their unconjugated forms, indicating that their capability for adherence was less hindered. All three cyclic doxorubicin conjugates sank into the membrane with the cargo positioned inside the medium. In the case of the SFTI-1-doxorubicin-conjugate, both the peptide and the cargo part positioned close to one surface (Figure 9D), while in the case of MCoTI-II- and Kalata-B1-doxorubicin-conjugates, the peptide parts were located in the proximity of one membrane surface while the cargos were translocated towards the opposing surface (Figure 9E,F). In the case of the rasagiline conjugates, all three peptides sank into the membrane, but the position of the cargo was very different. With SFTI-1, the rasagiline positioned close to the surface (Figure 9G); with Kalata B1, the rasagiline moved towards the center of the bilayer (Figure 9H); and when it was conjugated with MCoTI-II, it was closer to the opposite surface (Figure 9I). Similarly, the peptide part of all three zidovudine conjugates also sank deeply into the bilayer, and, with both SFTI-1 and Kalata-B1, the cargo remained in the relative vicinity of the surface (Figure 9J,K). However, when conjugated with MCoTI-II, it positioned towards the direction of the opposing surface (Figure 9L). It is also important to point out that the simulation with MCoTI-II-zidovudine was the only example where a significant distortion of the membrane was observed, although neither full penetration nor perforation took place.

## 3. Discussion

The original aim of this study was to simulate the penetration of the CPPs and conjugates throughout the POPC membrane bilayer.

A complete membrane penetration was not observed in 1000 ns for any of the molecules investigated. Only penetratin showed a significant structural rearrangement during the simulation, as the mostly helical structure uncoiled and a double-stranded β-sheet-like structure connected with a turn was formed. During the process, the peptide tightly adhered to the surface of the membrane with the formation of a number of hydrogen bonds and salt bridges. Ionic interactions were observed between the positively charged arginine side chains of the cationic peptides and the negatively charged head groups of the membrane phospholipids. Phe-modified penetratin and dodeca-penetratin derivative showed different behavior as they maintained their original helical structures. Instead of laying on the membrane, both peptides sank partially into it, with their N-terminal of the helix partially merging into the lipid bilayer, while the greater portion of the peptides remained above the membrane. Fewer H-bonds and salt bridges were formed compared to penetratin, but some additional π-cationic interactions were also observed.

The three cyclic peptides behaved in a completely different manner: they extruded the water between themselves and the membrane, and they were more tightly fitted to the lipid bilayer forming direct interactions. The attachment of the peptide to the membrane can be explained by entropic reasons, with water exclusion as the main cause.

In general, the conjugated molecules did not interact with the CPPs during the simulation. However, the conjugation of the drug molecule, in some cases, influenced interacting behavior between the membrane and the molecules.

Doxorubicin is an amphiphilic molecule possessing a hydrophobic anthraquinone ring substituted with a hydrophilic aminosugar derivative. When doxorubicin was attached to penetratin, the conjugate diverged from the membrane. In contrast, the Phe-derivative–doxorubicin conjugate behaved in a different way. Namely, the N-terminus equipped with the conjugate merged slightly into the membrane. The dodeca-penetratin derivative–doxorubicin conjugate merged slightly into the membrane with its C-terminal. The hydrophylic part of the molecule can form hydrogen bonds with phospholipids. These facts suggest that the amphipathic nature of doxorubicin influences the behavior of the conjugate.

The next drug investigated was zidovudine, which is a more hydrophilic molecule in comparison to doxorubicin. When it was attached to penetratins, no interaction was found between the two parts of the conjugates. The C-terminal of the penetratin conjugate slightly merged into the membrane, while the N-terminal with the zidovudine remained in the water. In the case of the Phe-derivative–zidovudine conjugate, the assembly diverged from the membrane and persisted between water molecules. In the case of the 12 AA-long dodeca-penetratin derivative conjugate, in turn, the N-terminus slightly merged into the membrane. The phosphate group of the zidovdine was able to form a salt bridge to the choline part of a POPC molecule.

Rasagiline, a hydrophobic compound, was also tested, and it conjugated to penetratins. All three compounds behaved in the same way, with the peptidic part retaining its helical conformation and merging slightly into the membrane with their N-terminal part. However, the rasagiline part merged deeply into the bilayer because of its highly nonpolar nature. The aromatic ring can form a π-cation interaction with the choline part of lecithine.

For the cyclic CPPs, the polarity of the small organic molecule had a dominant influence with respect to the behavior of the conjugate. In all cases, the peptide part was attached to the membrane, and water was extruded. Doxorubicin, as a conjugate, slightly merged into the membrane and formed a hydrogen bond, with the head part of the lipid oxygen atom bound to the phosphorous atom. Zidovudine diverged from the membrane because of its hydrophilic nature and formed hydrogen bonds with water molecules. The most hydrophobic rasagiline deeply merged into the membrane as long as its linker allowed.

## 4. Methods

### 4.1. Preparation of Peptides and Conjugates

The graphical user interface (GUI) Schrödinger molecular modeling package Maestro was used in the process of this study (Schrödinger Release 2022-3: Maestro, Schrödinger, LLC, New York, NY, USA, 2022). The peptide structures were downloaded from Research Collaboratory for Structural Bioinformatics Protein Database (RCSB PDB, http://rcsb.org accessed on 16 June 2023) based on the identifying code (PDB ID); see Table 1 [62]. All entries were derived from NMR spectroscopy, with multiple structures always working with the first member of the ensemble. Each structure was prepared using Schrödinger Protein Preparation Wizard, and the preprocess option was used to cap the termini of the linear peptides (the N-terminal was acetylated, and the C-terminal was transformed into an N-methyl-amide group) [63]. (Schrödinger Release 2022-3: Protein Preparation Wizard; Epik, Schrödinger, LLC, New York, NY, USA, 2022; Impact, Schrödinger, LLC, New York, NY, USA; Prime, Schrödinger, LLC, New York, NY, USA, 2022).

Drug molecules were drawn by the sketcher of the Maestro GUI, and they were minimized using the LigPrep module (Schrödinger Release 2022-3: LigPrep, Schrödinger, LLC, New York, NY, USA, 2022). The conjugates were made by merging the optimized small-molecule and selected peptide structures, and the linkers were added using the 3D Builder application within the interface: a glutaryl group for doxorubicin, a triazole ring for rasagiline, and phosphoric amide for zidovudine as indicated in the literature. In the case of the three linear peptides, the conjugations were formed through their N-terminal, while an appropriate amino acid side chain was utilized in the case of the cyclic peptides (Lys5 for SFTI-1, Lys14 for MCoTI-II, and Thr4 for Kalata B1; Table 4) [10,11,12]. That is, the simulations included the three unconjugated peptides together with all possible combinations of the three drugs and the six peptides, resulting in the building of 24 different peptides and conjugates.

### 4.2. Setup and Building of the Systems

All MD simulations in this study were completed with the Desmond Molecular Dynamic software under Schrödinger (Schrödinger Release 2022-3: Desmond Molecular Dynamics System, D. E. Shaw Research, New York, NY, USA, 2022. Maestro–Desmond Interoperability Tools, Schrödinger, New York, NY, USA, 2022). Setups for the runs were assembled with Desmond System Builder application under Maestro. All simulations were run within an orthorhombic box full of explicit water molecules generated by the single-point charge (SPC) model [46]. This enclosed a unimolecular membrane bilayer made of palmitoyl-oleoyl-phosphatidylcholine (POPC) compounds added automatically. The peptides and conjugates were manually placed on the top of the membrane at a distance of approximately 10 Å from its surface. The size of the membrane was calculated by the software using the buffer method, with a medium spread of 10 Å in every direction from the peptide or conjugate [64]. The assembling was continued with water boxes on both the top and bottom of the membrane, also according to the buffer. The assembly was then completed with sodium and chloride ions to statistically reach the isotonic (0.15 M) concentration, and additional counter ions were added if needed in order to neutralize the charge of the peptide or conjugate so the net charge of the system was reduced to zero. Prior to the MD simulations, the assemblies were minimized with OPLS3e (optimized potential for liquid simulations) force field method for the final positioning of the molecules to avoid steric clashes. The OPLS-AA is an all-atom force field parameter for both proteins and many general classes of organic molecules; therefore, no further parametrization for the drug molecules is necessary [65,66,67]. According to the literature, this system is suitable for modeling peptides in the presence of POPC membrane [68].

All 24 peptides and peptide conjugates mentioned above were placed into the simulation box with POPC membrane; that is, altogether, 24 systems were included in the study.

### 4.3. MD Simulations

The completed setups were then loaded into Desmond’s Molecular Dynamics interface, and simulations for 1000 ns runs were initialized [67]. All MD simulations began with the standard relaxation protocol, which also included equilibration utilizing the default settings: starting with 12 ps-long NVT (constant substance, volume, and temperature) ensemble simulation at 10 K temperature; followed by two 12 ps-long NPT (constant substance, temperature, and pressure) ensemble simulation at 10 K temperature and 1.01325 bar pressure; and, finally, a 24 ps-long NPT ensemble simulation at 300 K temperature and 1.01325 bar pressure [50].

Following the relaxation, all MD simulations were always carried out with NPT settings, where the pressure was 1.01325 bar, and the temperature was 300 K [69]. Additionally, the recording interval of the trajectory was set to 1000 ps (therefore, each trajectory contained 1000 frames).

Simulations were completed on hardware with nVidia^®^ GeForce GTX 1070 Ti 1683 MHz x2432 graphics processing unit (GPU) under Linux Ubuntu. The 1000 ns simulations took a maximum of 70–80 h.

### 4.4. Analyzing the Structures

The Structure Analysis application of the Schrödinger was used to evaluate the trajectories (Schrödinger Release 2022-3: Prime, Schrödinger, LLC, New York, NY, USA, 2022). The most critical piece of data is the root mean square deviation (RMSD) of the alpha carbon atoms depending on the running time. All RMSD values were calculated compared to the 0 ns geometry of the trajectories (after relaxation/equilibration). If the RMSD does not change, the conformation is stable. In contrast, if the RMSD increases or decreases, the atoms are moving, and the system is not in equilibrium [70,71,72].

The evaluation also included monitoring the change of the number of intramolecular hydrogen bonds within the peptides over time, which correlates with the changes of the secondary structure. Fewer intramolecular hydrogen bonds may be indicative of irregular, less stable conformations, while a higher number usually means a more organized and energetically more stable folded structure.

The comparison of the Ramachandran plots of the peptides at different times of the simulation can also show structural differences. In the case of minor conformational movements during simulation, the Φ and Ψ dihedrals were quite similar at the initial and final frames. However, where significant changes were observed in the Φ and Ψ dihedrals on the Ramachandran plot, the coordinates of the dominant conformations changed.

Because of the limitations of the software, we were unable to track all interactions between the peptides, small molecules, and the membrane over the course of the simulations. Therefore, we counted the hydrogen bonds, salt bridges, and π-cation interactions marked by the graphical interface manually in the final frame (at 1000 ns) within each trajectory. The entire modeling process has been summarized on a working flowchart (see Figure 10).

## 5. Conclusions

In this study, we examined the behavior of CPPs with covalently conjugated drug molecules using all-atom MD simulations. Although a complete membrane penetration was not achieved, some interesting conformational and positional changes were observed during the 1000 ns simulation time.

We found that only the unconjugated penetratin underwent some major conformational rearrangement, while less flexible 6,14-Phe-penetratin and dodeca-penetratin retained their mostly helical structure. Penetratin and analogues thereof were more affected by the polarity of the conjugated small molecule. Namely, the hydrophilic zidovudine seemingly inhibited the interaction between the peptide and the membrane, the more hydrophobic rasagiline guided the entire conjugate in between the membrane bilayer, and the amphiphilic doxorubicin induced variable degrees of penetration for each peptide.

The three cyclic peptides (SFTI-1, Kalata B1, and MCoTI-II) behaved in a similar manner during the simulations. Due to their high structural stability, only minimal conformational changes were observed, and their position compared to the surface of the lipid bilayer was altered less. The influence of the conjugates for the penetration also seemed to be less significant, but the conjugated small molecules were oriented according to their polarity.

The lack of direct penetration might be the result of the relative simplicity of the model. Although we used all-atom MD, a simple monomolecular membrane model was applied, and neither the possible membrane components nor the membrane potential could be implemented properly. Our system contained only a single CPP peptide and, therefore, more complex multi-molecular mechanisms, such as complexation with other proteins or pore formation, could not be examined.

It is our sincere hope that we will be able to build a much larger simulation box, including more than just a single peptide in a sufficiently high concentration, and that, consequently, spontaneous penetration may be observed.

## Figures and Tables

**Figure 1 pharmaceuticals-16-01251-f001:**
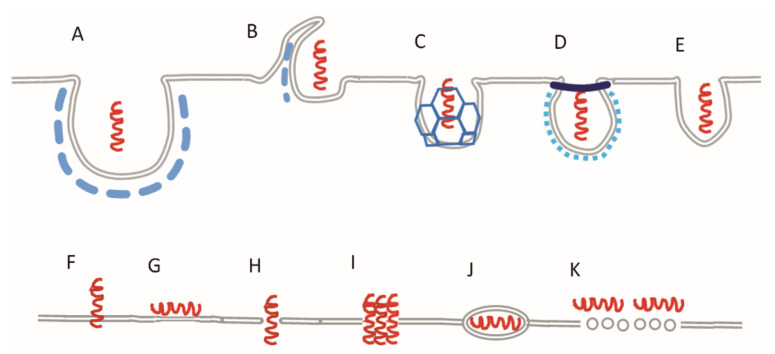
Active (**A**–**E**) and passive (**F**–**K**) mechanisms of the CPP penetration: (**A**) phagocytosis; (**B**) macropinocytosis; (**C**) clathrin-mediated endocytosis; (**D**) caveola; (**E**) constitutive endocytosis; (**F**) direct translocation; (**G**) membrane thinning; (**H**) toroidal pore; (**I**) barrel pore; (**J**) inverted micelle; (**K**) carpet model.

**Figure 2 pharmaceuticals-16-01251-f002:**
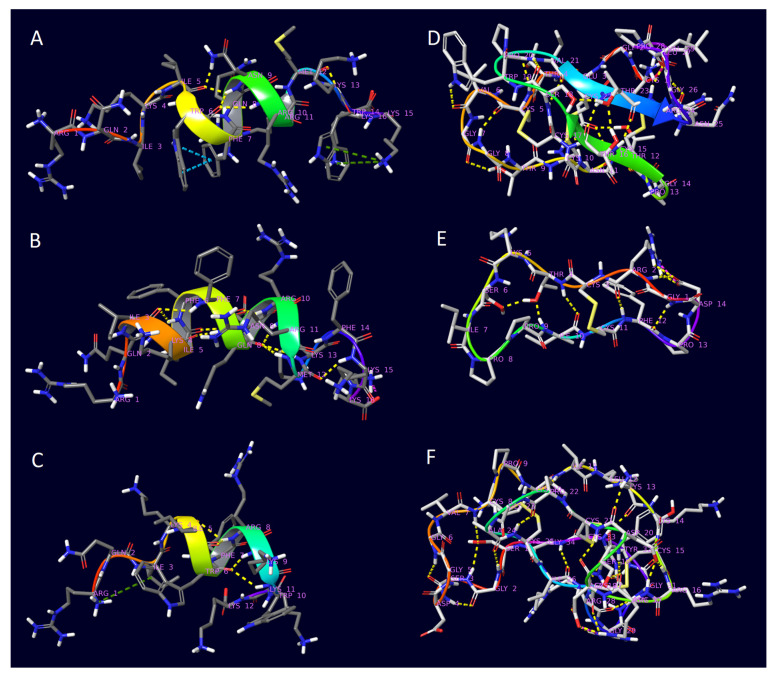
The 3D structures of the peptides determined by NMR from the Protein Database by the following PDB IDs: (**A**) 1KZ0—penetratin; (**B**) 1KZ2—6,14-Phe-penetratin; (**C**) 1KZ5—dodeca-penetratin; (**D**) 1NB1—Kalata B1; (**E**) 1JBL—SFTI-1; (**F**) 1HA9—MCoTI-II. Residue positions are colored from red to violet and intramolecular interactions are represented as dashed lines: hydrogen bond—yellow; π–π stacking—turquoise; π-cationic—dark green; salt bridge—purple.

**Figure 3 pharmaceuticals-16-01251-f003:**
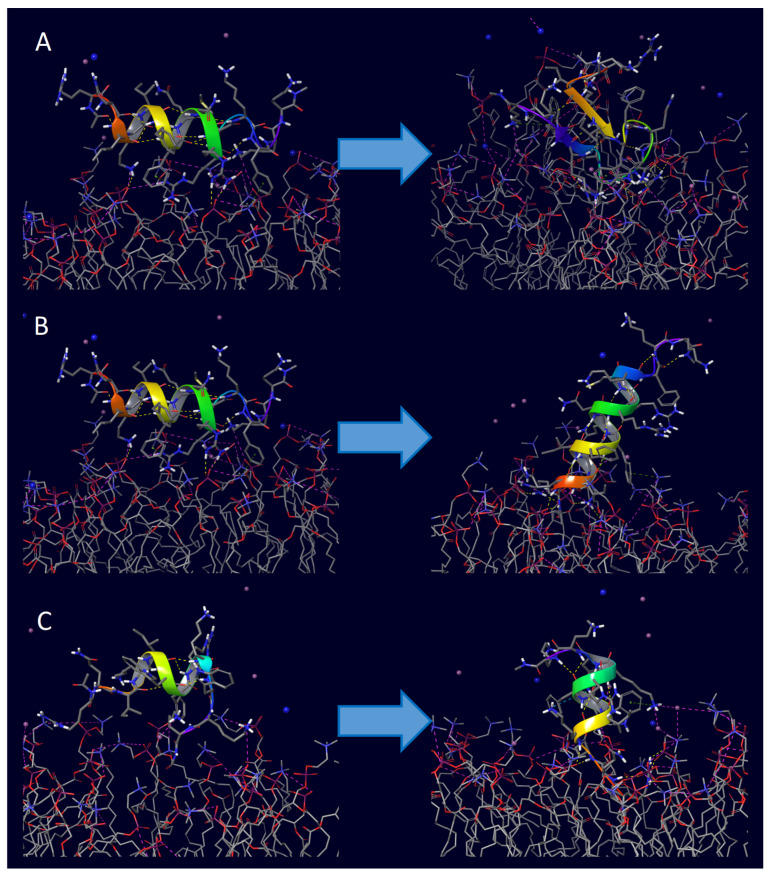
Comparison of the initial (**left**) and final (**right**) positions of unconjugated CCPs during the 1000 ns membrane simulations: (**A**) penetratin with POPC; (**B**) 6,14-Phe-penetratin with POPC; (**C**) dodeca-penetratin with POPC—all peptides starting from the surface of bilayer.

**Figure 4 pharmaceuticals-16-01251-f004:**
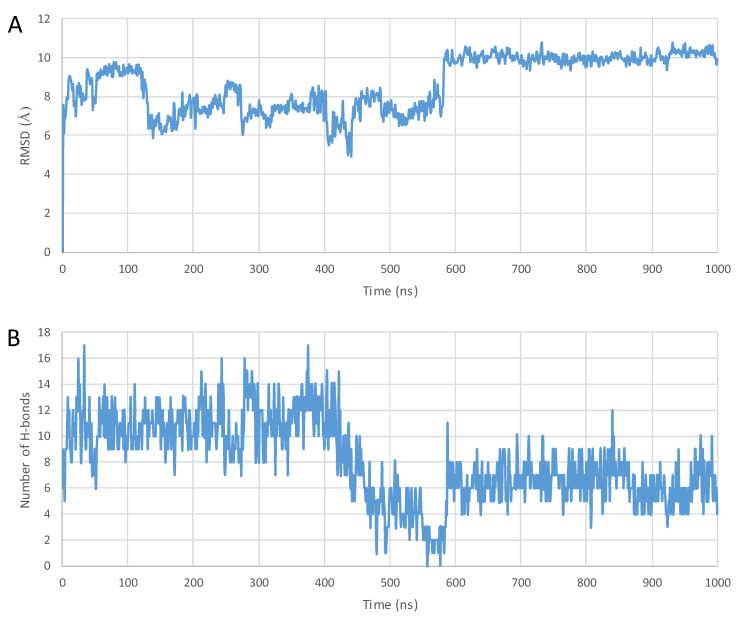
(**A**) RMSD diagram of the α-carbon atoms of the unconjugated penetratin peptide; (**B**) the total number of intramolecular hydrogen bonds plotted against simulation time during the 1000 ns MD simulation—starting from the top of the membrane bilayer.

**Figure 5 pharmaceuticals-16-01251-f005:**
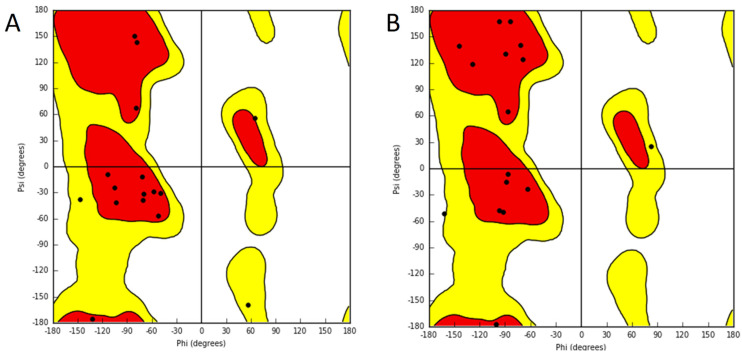
Comparison of the Ramachandran plots of the unconjugated penetratin peptide (**A**) at the beginning; (**B**) at the end of the 1000 ns simulation with the POPC membrane model—starting from the top of the membrane bilayer.

**Figure 6 pharmaceuticals-16-01251-f006:**
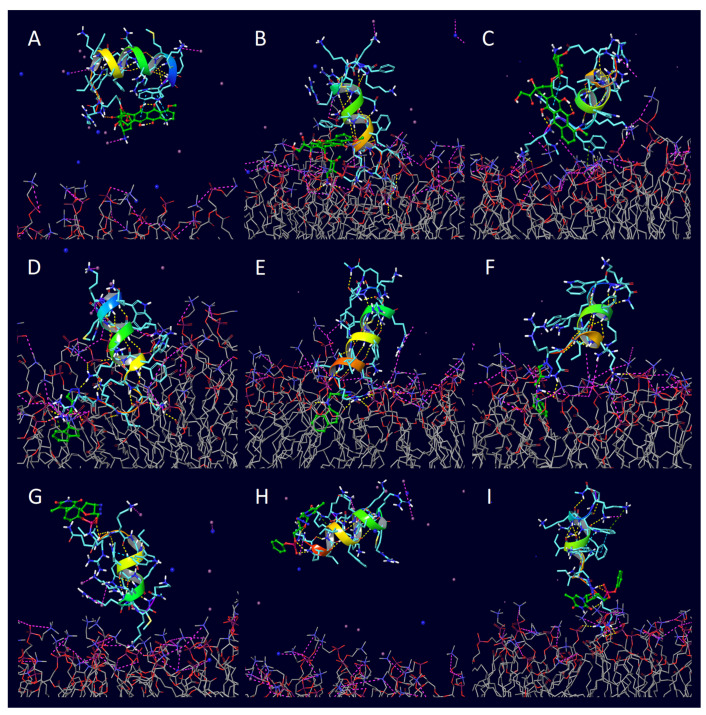
The final positions of the 1000 ns POPC membrane simulations with different CCP-conjugates: (**A**) penetratin–doxorubicin, (**B**) 6,14-Phe-penetratin–doxorubicin, (**C**) dodeca-penetratin–doxorubicin, (**D**) penetratin–rasagiline, (**E**) 6,14-Phe-penetratin–rasagiline, (**F**) dodeca-penetratin–rasagiline, (**G**) penetratin–zidovudine, (**H**) 6,14-Phe-penetratin–zidovudine, (**I**) dodeca-penetratin–zidovudine—all conjugates were started from the top of the membrane bilayer.

**Figure 7 pharmaceuticals-16-01251-f007:**
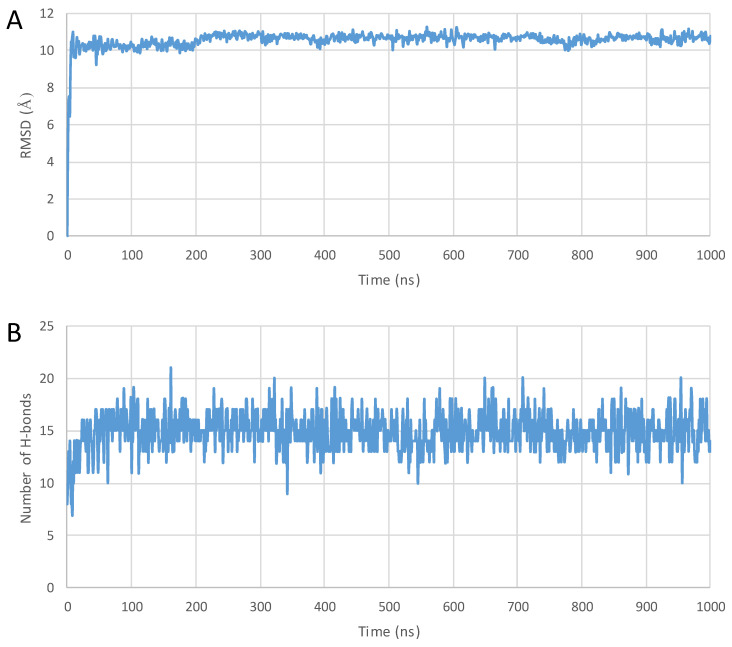
(**A**) The Root Mean Square Deviation (RMSD) diagram of the α-carbon atoms of the peptide; (**B**) the total number of intramolecular hydrogen bonds plotted against simulation time during the 1000 ns MD simulation of penetratin–doxorubicin conjugate—starting from the top of the membrane bilayer.

**Figure 8 pharmaceuticals-16-01251-f008:**
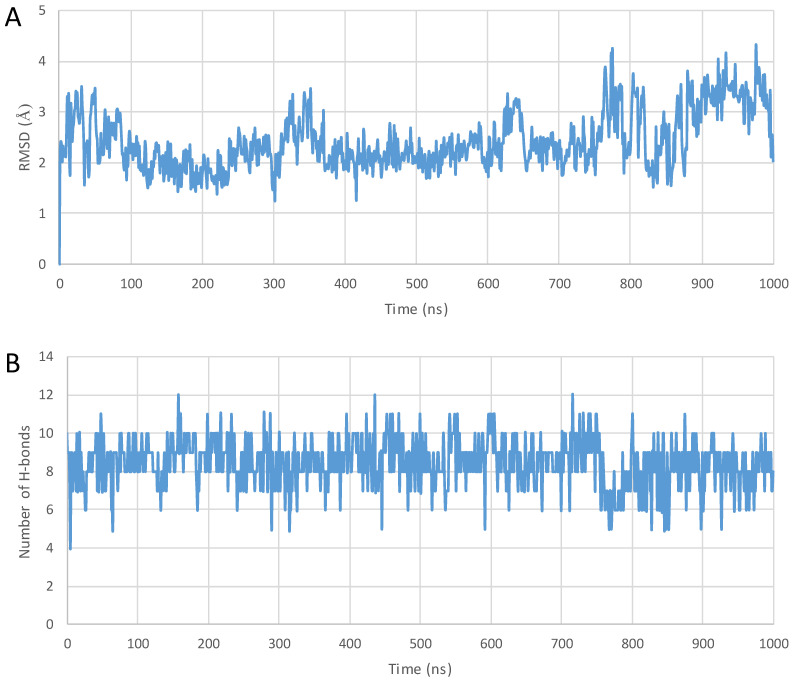
(**A**) The Root Mean Square Deviation (RMSD) diagram of the α-carbon atoms of the peptides; (**B**) the total number of intramolecular hydrogen bonds plotted against simulation time during the 1000 ns MD run of the unconjugated SFTI-1 peptide—starting from the top of the membrane bilayer.

**Figure 9 pharmaceuticals-16-01251-f009:**
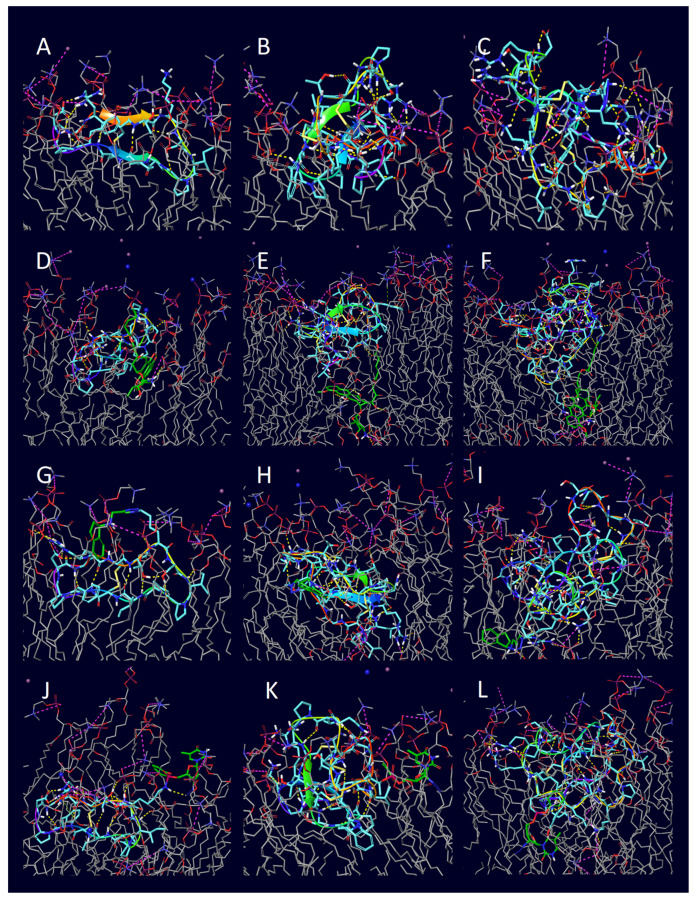
The final positions of the 1000 ns POPC membrane simulations with cyclic CCPs and their conjugates: (**A**) unconjugated SFTI-1, (**B**) unconjugated Kalata B1, (**C**) unconjugated MCoTI-II, (**D**) SFTI-1-doxorubicin, (**E**) Kalata B1-doxorubicin, (**F**) MCoTI-II-doxorubicin, (**G**) SFTI-1-rasagiline, (**H**) Kalata B1-rasagiline, (**I**) MCoTI-II-rasagiline, (**J**) SFTI-1-zidovudine, (**K**) Kalata B1-zidovudine, (**L**) MCoTI-II-zidovudine—all conjugates were started from the top of the membrane bilayer.

**Figure 10 pharmaceuticals-16-01251-f010:**
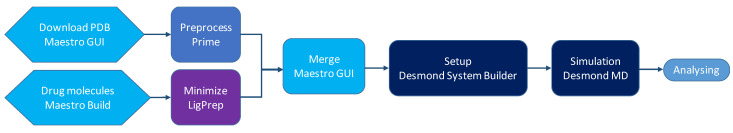
Schematic chart of the workflow.

**Table 1 pharmaceuticals-16-01251-t001:** Amino acid sequences of the investigated peptides.

Peptide	PDB ID	Sequence	Reference
Penetratin	1KZ0	RQIKIWFQNRRMKWKK	[52]
6,14-Phe-penetratin	1KZ2	RQIKIFFQNRRMKFKK	[52]
Dodeca-penetratin	1KZ5	RQIKIWFRKWKK	[52]
Kalata B1	1NB1	[CGETCVGGTCNTPGCTCSWPVCTRNGLPV]	[53]
SFTI-1	1JBL	[GRCTKSIPPICFPD]	[54]
MCoTI-II	1HA9	[SGSDGGVCPKILKKCRRDSDCPGACICRGNGYCG]	[55]

**Table 3 pharmaceuticals-16-01251-t003:** The number of observed interactions between the peptides/conjugates and the POPC membrane molecules at the end of the 1000 ns simulations.

Entry	Peptide	Conjugate	H-Bond	π-Cation	Salt Bridge
1	penetratin (1KZ0)	unconjugated	11	0	4
2	penetratin (1KZ0)	doxorubicin	0	0	0
3	penetratin (1KZ0)	rasagiline	12	1	7
4	penetratin (1KZ0)	zidovudine	1	0	2
5	6,14-Phe-penetratin (1KZ2)	unconjugated	7	1	5
6	6,14-Phe-penetratin (1KZ2)	doxorubicin	8	0	3
7	6,14-Phe-penetratin (1KZ2)	rasagiline	10	0	4
8	6,14-Phe-penetratin (1KZ2)	zidovudine	0	0	0
9	dodeca-penetratin (1KZ5)	unconjugated	6	0	1
10	dodeca-penetratin (1KZ5)	doxorubicin	2	1	3
11	dodeca-penetratin (1KZ5)	rasagiline	5	0	4
12	dodeca-penetratin (1KZ5)	zidovudine	4	0	2
13	SFTI-1 (1NB1)	unconjugated	0	0	0
14	SFTI-1 (1NB1)	doxorubicin	10	0	2
15	SFTI-1 (1NB1)	rasagiline	6	0	2
16	SFTI-1 (1NB1)	zidovudine	2	0	2
17	Kalata B1 (1JBL)	unconjugated	2	0	0
18	Kalata B1 (1JBL)	doxorubicin	6	0	3
19	Kalata B1 (1JBL)	rasagiline	4	0	3
20	Kalata B1 (1JBL)	zidovudine	3	0	1
21	MCoTI-II (1HA9)	unconjugated	0	0	0
22	MCoTI-II (1HA9)	doxorubicin	8	0	11
23	MCoTI-II (1HA9)	rasagiline	6	0	8
24	MCoTI-II (1HA9)	zidovudine	16	0	6

**Table 4 pharmaceuticals-16-01251-t004:** Schematic representation of conjugates investigated.

	Doxorubicin	Rasagiline	Zidovudine
**penetratin and analogues**	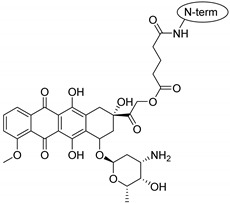	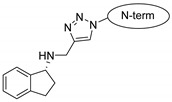	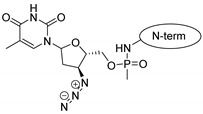
**Kalata B1**	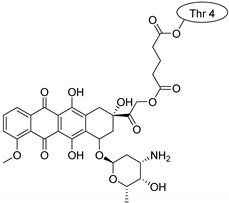	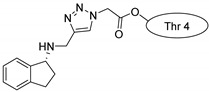	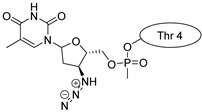
**SFTI-1**	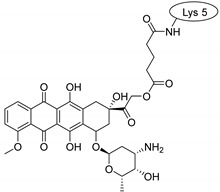	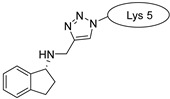	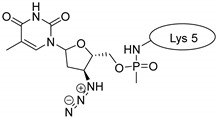
**MCoTI-II**	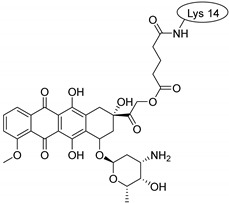	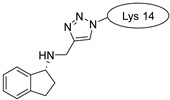	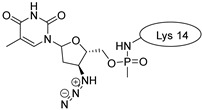

## Data Availability

Data is contained within the article and Appendix A.

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
