# Peer review of "Molecular Dynamics Simulations of Drug-Conjugated Cell-Penetrating Peptides"

_pharmaceuticals, 2023, doi:10.3390/ph16091251_

Round 1

Reviewer 1 Report

In the current work, all-atom simulations were carried out to understand the interactions of the cell-penetrating peptides and their conjugates with the membrane. The topic is attractive and the authors have presented some interesting results. However, I do not recommend publish this paper in the current version for three concerns:

1. In the third paragraph of Page 2, the authors state that “Our aim is to solve these problems with molecular dynamics simulations through a number of examples.” However, the authors did not put forward the problems they want to solve before this sentence. This is a major limitation of the Introduction part, the Abstract or the whole work. In fact, it’s of critical importance to let readers know in a straightforward way for what purpose the authors want to carry out these simulations.

2. Another concern is that the authors did not replicate their simulations for any system, which makes the conclusion less reliable. Tanking Figure 2 as an example, the orientation of these helices relative to the membrane could be caused by some random factors or insufficient sampling. Therefore, the authors should introduce replication at least for some systems in this work. The reviewers also think that the authors put too many systems in one paper. In fact, this paper can be divided into two or more papers, with each system running with enough replications and with more in-depth analysis.

3. The authors reported that the interaction of penetratin with the membrane led to spectacular rearrangements in the secondary structure of the peptide”. In fact, it’s not safe to get this conclusion, if they did not simulate the peptide in a solvent without presence the lipid membrane.

Some minor errors:

The citation should always be placed before the period. Please check throughout the manuscript and make corrections.

Page 3, paragraph 3, line 3: please delete the word “again” here.

Page 3, paragraph 3, line 3: hydrophyllic hydrophilic

Page 3, paragraph 4:This paragraph is not written clearly. Is it about a general principle, or just for the specific case?

Page 3, the last paragraph, the last line but 5: “we intend to investigate the elimination of …” should be changed to: we intend to investigate whether the elimination of ….

Page 5, paragraph 1, line 3: “peptid conjugatespeptide conjugates

Figure 7: include some unnecessary part by mistake.

Author Response

Answers to reviewer 1.

We are very grateful for the careful reading and reviewing of the manuscript. Our answers are detailed below.

We kindly ask you to accept our answers for the questions and comments.

In the current work, all-atom simulations were carried out to understand the interactions of the cell-penetrating peptides and their conjugates with the membrane. The topic is attractive and the authors have presented some interesting results. However, I do not recommend publish this paper in the current version for three concerns:

  1. In the third paragraph of Page 2, the authors state that “Our aim is to solve these problems with molecular dynamics simulations through a number of examples.” However, the authors did not put forward the problems they want to solve before this sentence. This is a major limitation of the Introduction part, the Abstract or the whole work. In fact, it’s of critical importance to let readers know in a straightforward way for what purpose the authors want to carry out these simulations.

Answer: The original aim was to model the internalization process, but it did not occur. Therefore, we were focusing some aspects might be involved in the penetration instead, such as the formation of intermolecular interactions, orientation and conformational changes of the penetratin and analogues, and the influence of the conjugated small molecule on the process. Nevertheless, this study gives some insight might help better understanding the initialization of penetration. (The text has been modified accordingly, page 2. paragraph 3).

  1. Another concern is that the authors did not replicate their simulations for any system, which makes the conclusion less reliable. Tanking Figure 2 as an example, the orientation of these helices relative to the membrane could be caused by some random factors or insufficient sampling. Therefore, the authors should introduce replication at least for some systems in this work. The reviewers also think that the authors put too many systems in one paper. In fact, this paper can be divided into two or more papers, with each system running with enough replications and with more in-depth analysis.

Answer: We carried out follow-up simulations as the referee suggested with the dodeca-penetratin-rasagiline and SFTI-1-rasagiline conjugates. These runs were started from a slightly different positioning of the peptide and, also the orientations of the small molecules were altered while the general distance from the membrane was the similar and all conditions and settings were unchanged. At the end of the runs the positions and the interactions were compared with those of the original simulations, and we have found that these results were not exactly the same, but they were very similar. (Please, see attached document). The other suggestion of the referee was that we studied too many systems and we should divide into two or more papers is not possible at this point. This work is a closure of a PhD work and must summary several months of work. We have already left out several simulations, further reduction would make the comparison of the different systems impossible, we hope the reviewer will understand this.

  1. The authors reported that “the interaction of penetratin with the membrane led to spectacular rearrangements in the secondary structure of the peptide”. In fact, it’s not safe to get this conclusion, if they did not simulate the peptide in a solvent without presence the lipid membrane.

Answer: We have completed simulations with all the peptides and conjugates in water boxes too (Please, see attached document). As mentioned before, these results were part of a more extensive project, but – except for the native penetratine – the results in water boxes were not particularly interesting therefore we decided to leave these out because we had too many systems already. In contrast, the structure of the penetratine at the end of the simulation with membrane was very much different from the water box.

Comparison of penetratin in water box and on the POPC membrane model

Penetratin in water box

Penetratin on the surface of POPC membrane model

Ramachandran plot of penetratin in water box, starting and final frame

Ramachandran plot of penetratin on the surface of POPC membrane model, starting and final frame

Some minor errors:

The citation should always be placed before the period. Please check throughout the manuscript and make corrections. Answer: the citations have been corrected.

Page 3, paragraph 3, line 3: please delete the word “again” here.           Answer: corrected

Page 3, paragraph 3, line 3: hydrophyllic → hydrophilic             Answer: corrected.

Page 3, paragraph 4:This paragraph is not written clearly. Is it about a general principle, or just for the specific case?         Answer: this one meant for this specific case (described within refereve No. 46) and the text has been modified accordingly.

Page 3, the last paragraph, the last line but 5: “we intend to investigate the elimination of …” should be changed to: we intend to investigate whether the elimination of ….     Answer: it has been changed

Page 5, paragraph 1, line 3: “peptid conjugates” → peptide conjugates              Answer: corrected.

Figure 7: include some unnecessary part by mistake.
Answer: this was glitch occurred during the pdf generation. All figures were replaced by new, high-resolution versions as requested by another referee.

Reviewer 2 Report

Dear Authors,

I feel pleasure to review quality manuscript entitled; Molecular dynamics simulations of drug-conjugated cell-penetrating peptides which is significant work in the field of oncology to discover novel chemotherapeutic agents. Overall manuscripts fulfill the criteria of acceptance.

Some minor drawbacks are listed below:

1-                   Introduction part is too much large, authors should add flowchart in the introduction part like on mechanism of penetration (two main types of either energy-dependent or independent) and on translocation of CPP via different MD simulation methods.  

2-                  Authors should improves the presentation of figures throughout manuscript and minor English check is required throughout the manuscript.

3-                  What is rational design of selecting these three different class of drugs for this study?

a-                  Doxorubicin is a topoisomerase-2 inhibitor anticancer drug.

b-                  Rasagiline is a specific, irreversible MAO-B inhibitor used for the treatment of Parkinson’s disease.

c-                  Zidovudine, a reverse transcriptase inhibitor antiviral drug, has been developed to cure HIV infection

These three drugs are different in structures as well as function, what is rational to select these drugs against each peptide.

Authors should add rationale in the form figure (preferably) supported by literature.

4-                  Authors should add working flowchart figure which having details from beginning to outcomes of this study.

5-                  Authors should add outcomes of this study in the conclusion part of this manuscript.

Authors can consult these below mentioned papers to improve the manuscript.

1-                  https://doi.org/10.3390/molecules23020295

2-                  DOI: 10.1039/D1NA00674F

3-                  https://doi.org/10.1021/acs.bioconjchem.8b00208

4-                  https://doi.org/10.1002/ddr.22076

5-                  https://doi.org/10.1021/mp3004034

6-                  https://doi.org/10.3389/fcimb.2022.838259

Thanks and Regards

Reviewer

Author Response

Answers to reviewer 2.

We are very grateful for the careful reading and reviewing of the manuscript. Our answers are detailed below.

We kindly ask you to accept our answers for the questions and comments.

  1. Introduction part is too much large, authors should add flowchart in the introduction part like on mechanism of penetration (two main types of either energy-dependent or independent) and on translocation of CPP via different MD simulation methods.  

Answer: We have completed the manuscript with a new figure (Figure 1.) summarizing the mechanisms of penetration. Similar figures were found in an article recommended by the referee.

  1. Authors should improves the presentation of figures throughout manuscript and minor English check is required throughout the manuscript.

Answer: We re-exported all images from the modelling software and re-edited them into new high resolution (300 dpi) figures as requested. The text also has been revised and several corrections were made in order to improve quality.

  1. What is rational design of selecting these three different class of drugs for this study?
    1. Doxorubicin is a topoisomerase-2 inhibitor anticancer drug.
    2. Rasagiline is a specific, irreversible MAO-B inhibitor used for the treatment of Parkinson’s disease.
    3. Zidovudine, a reverse transcriptase inhibitor antiviral drug, has been developed to cure HIV infection

These three drugs are different in structures as well as function, what is rational to select these drugs against each peptide.

Authors should add rationale in the form figure (preferably) supported by literature.

Answer: We intend to investigate that if the cargo have any influence on the penetration, therefore we decided to include three compounds with different physicochemical properties, as stated in the introduction (on page seven and also in Table 1). The reason we combined them with each peptides was that we wanted see if any of the peptides is more capable for transferring cargo than the other. Only doxorubicin among the three compounds were investigated as CPP cargo (as mentioned in one study recommended by the referee – this is already included in the article, see Reference 11). As also mentioned in the Introduction (Page 7), Rasagiline has to get through the blood–brain barrier while in case of Zidovudine conjugation can increase its specificity towards the infected cells, thereby reducing the side effects.   We believe that conjugation with CPPs could enhance these and if biological assays would be completed in the future we also would be able to test which of these peptides performs better.

  1. Authors should add working flowchart figure which having details from beginning to outcomes of this study.

Answer: We have added a working flowchart (as Figure 3) at the end of the Methods as requested by the referee.

  1. Authors should add outcomes of this study in the conclusion part of this manuscript.

Authors can consult these below mentioned papers to improve the manuscript.

  • https://doi.org/10.3390/molecules23020295
  • https://doi.org/10.1039/D1NA00674F
  • https://doi.org/10.1021/acs.bioconjchem.8b00208
  • https://doi.org/10.1002/ddr.22076
  • https://doi.org/10.1021/mp3004034
  • https://doi.org/10.3389/fcimb.2022.838259

Answer: We thank the referee for his/her kind help. We have carefully examined the recommended references, some of these were already cited in this article but now we have included them all. We hope that you will find it acceptable with these improvements.

Reviewer 3 Report

The authors identified the better approach for this study.  However, the study needs better revision to proceed further.  From the present form, the manuscript cannot proceed further. Hence, I consider a Major Revision, and the comments are given below.

Fig 1,2  is not clear.  Hence, author should use appropriate software to draw the image with 300dpi and it should be completely replaced (check the instruction to the authors).

The structure size is not uniform in the Table 3.  It has to be redrawn. 

The manuscript should be written towards the publication quality.  To check flow of manuscript, English Language etc.,

Page Number 9 off 22: ...Because of the limitations of the software we were unable to track all interactions between the peptides, small molecules and the membrane in the course of the whole simulations.

Authors can perform the simulated interaction diagram analysis to examine the protein-ligand contacts, hydrogen bond interctions etc.,

All the RMSD graphs can be overlaid for the better understanding.

What is the reason for solvating it into the POPC bilayer membrane

What is the size of the box?

English language must be improved

Author Response

Answers to reviewer 3.

We are very grateful for the careful reading and reviewing of the manuscript. Our answers are detailed below.

We kindly ask you to accept our answers for the questions and comments.

Fig 1,2 is not clear.  Hence, author should use appropriate software to draw the image with 300dpi and it should be completely replaced (check the instruction to the authors).

Answer: We re-exported all images from the modelling software and re-edited them into new high resolution (300 dpi) figures as requested.

The structure size is not uniform in the Table 3.  It has to be redrawn. 

Answer: We redrawn table as requested, now all the structures are sized and formatted uniformly according to the ACS template.

The manuscript should be written towards the publication quality.  To check flow of manuscript, English Language etc.,

Answer: The text also has been revised and several corrections were made in order to improve quality. The language of the article was also checked by an English language lecturer.

Page Number 9 off 22: ...Because of the limitations of the software we were unable to track all interactions between the peptides, small molecules and the membrane in the course of the whole simulations. Authors can perform the simulated interaction diagram analysis to examine the protein-ligand contacts, hydrogen bond interctions etc.,

Answer: Unfortunately, there is no tool in this software modelling package to specify two custom groups of atoms therefor we were unable count interactions between the membrane and conjugate without including all interactions among the membrane molecules within the trajectory.

All the RMSD graphs can be overlaid for the better understanding.

Answer: 24 systems were investigated altogether in this study, not all of them were included into the final version, but all the RMSD graphs can be found in either in the text or in the supplement. Overlaying them all in a single graph would be overcrowded and uninformative, plus due different amplitudes the scaling of the y axis is different in some of the runs.

What is the reason for solvating it into the POPC bilayer membrane

Answer: POPC is the main component of biological membranes and monocomponent POPC bilayers and it has been extendedly use in MD studies in the literature as membrane model. In one article particular POPC was utilized in a series of different types of molecular dynamics simulations including the OPLS force field, see Kurki et al., J. Chem. Inf. Model. 2022, 62, 6462-6474. DOI: 10.1021/acs.jcim.2c00395

What is the size of the box?

Answer: In this software the size of the box is not fixed (if there was a membrane in the system), the buffer method was used in which the box is scaled to the peptide with 10 Å buffer in each directions (this was precisely described in the supplement). The size of the boxes was:

  • 93 Å tall, 46 Å wide and 51 Å deep in case of the penetratin conjugates,
  • 94 Å tall, 34 Å wide and 38 Å deep in case of the SFTI-1 conjugates,
  • 92 Å tall, 35 Å wide and 52 Å deep in case of the Kalata B1 conjugates,
  • and 92 Å tall, 41 Å wide and 60 Å deep in case MCoTI-II conjugates, approximately.

Round 2

Reviewer 1 Report

The authors have solved the reviewer's concerns in the current version.

Author Response

Answer to Reviewer1:

Many thanks for accepting our answers and corrections.

Reviewer 3 Report

Figures must be improved 

1.  Refer and cite the following paper to present the figures in the manuscript and also to include the relevant MDS methodology

https://doi.org/10.3389/fchem.2023.1090630

2.  The figure quality especially fig 1 is not convinced for the publication

3.  Authors are advised to include the figure number in the appropriate form (follow the journal guidelines).  author mentioned figure x.x

Authors have improved the quality of English.

Author Response

Answer to Reviewer3:

  1. The paper suggested by the reviewer is cited now and the relevant MDS methodology was included
  2. The quality of Figure 1 was increased.
  3. The figure numbers are included in the appropriate form.

Many thanks for the comments and we ask the reviewer to accept our answers and corrections.

Round 3

Reviewer 3 Report

Authors had applied their effort to produce this outcome. Anyhow, it require some more.

English is Ok.